# Caffeic Acid-layered Double Hydroxide Hybrid: A New Raw Material for Cosmetic Applications

**Maria Bastianini \* 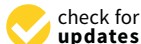, Caterina Faffa, Michele Sisani and Annarita Petracci**

R&D Department, Prolabin & Tefarm S.r.l., Via dell'Acciaio 9, 06134 Perugia, Italy;
caterina.faffa@gmail.com (C.F.); michele.sisani@prolabintefarm.com (M.S.);
annarita.petracci@prolabintefarm.com (A.P.)

**\*** Correspondence: maria.bastianini@prolabintefarm.com; Tel.: +39-328-714-2344; Fax: +39-075-591-9493

**Abstract:** Bioactive ingredients from natural sources possess well-known positive effects in cosmetic applications. Among them, phenolic acids have emerged with very interesting potential. Caffeic acid (CAF) is one of the most promising active compounds because it possess antioxidant, anti-inflammatory, antitumoral and anti-wrinkle effects. In order to increase its local bioavailability in topical applications, the vehiculation of caffeic acid can lead to a new raw material of cosmetic interest. For this purpose, clay minerals possess excellent properties, such as low or null toxicity and good biocompatibility. Clays are able to host a wide range of active ingredients in the interlayer region, using a green process known as intercalation reaction. The hosting of cosmetic actives into the layered structure of anionic clays allows the preparation of new materials with enhanced stability towards oxidation and photodegradation, better local bioavailability, and easier workability. In this paper, the successful vehiculation of caffeic acid into anionic clay is presented. The obtained hybrid is very promising for the cosmetic market because of its higher bioavailability and prolonged antioxidant activity.

**Keywords:** caffeic acid; layered double hydroxides; hybrids; anti-wrinkle; bioavailability; raw material

## 1. Introduction

Like other hydroxycinnamic acids, caffeic acid (3,4-Dihydroxycinnamic acid, Figure 1) is a powerful antioxidant. This polyphenol can be found in many food sources, including coffee drinks, blueberries, apples and cider [1].

Its activity seems to be strictly connected to the presence of the two hydroxyl groups on its aromatic ring, which is able to donate the hydrogen and stabilize the resulting phenoxylic radical [2].

Other mechanisms of action for this acid were also suggested, such as the ability to chelate divalent metals, typical catalysts of oxidation reactions and cofactors of enzymes responsible for the creation of reactive oxygen species, and the modulation of gene expression in an immunoregulatory and anti-inflammatory sense [2].

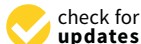

**Figure 1.** Chemical structure of caffeic acid.

The activity of caffeic acid is connected to the inhibition of tumor proliferation [3,4] and to the protection of low-density lipoproteins and their constituents from oxidation, such as vitamin E and phosphatidilcholine (hence the anti-inflammatory and antiatherosclerotic role) [5–7]. This species is also able to give radical [8] and singlet oxygen [9] extinction.

In cosmetics, caffeic acid has been demonstrated as a very effective active ingredient in the treatment of dermal diseases [10]. Furthermore, it has been shown to have anti-wrinkle activity in vivo [11].

The aim of this work is to vehiculate caffeic acid in order to develop a new effective ingredient with potential applications in the cosmetic market [12].

In the field of drug carriers, layered double hydroxides (LDHs) are promising materials due to their high active loading capacity. LDHs are perfectly biocompatible [13] clays that are already used in medicine as antiacid and have found many industrial applications for the production of a wide range of cosmetics, nutraceuticals, and pharmaceuticals [14,15]. LDHs possess a layered structure—its structure derives from that of the mineral brucite $Mg(OH)_2$, which has layers constituting the edge concatenation of different magnesium hydroxide octahedra. After a partial isomorphous substitution of the divalent cations by trivalent cations, an excess of positive charge arises on the layers that is counter-balanced by anions located in the interlayer region. LDHs are quite easy to prepare in the laboratory and have the general formula $[M(II)_{1-x}M(III)_x(OH)_2]^{x+}[A^{n-}_{x/n}]^{x-}\cdot nH_2O$, where M(II) is a divalent metal cation, usually Mg or Zn; M(III) is a trivalent metal cation, usually Al; x is the M(III)/[M(II) + M(III)] molar ratio, which has values generally ranging between 0.2 to 0.4 and which determines the positive layer charge density of the material; $A^{n-}$ is an exchangeable inorganic or organic anion that compensates for the positive charge of the layer; and n is the number of moles of water [16]. Due to the presence of the exchangeable anions ($A^{n-}$) in the interlayer region, LDHs are able to create a host-guest structure through an ion-exchange (intercalation) process. Intercalation reactions provide routes for the synthesis of new solids with chemical properties that can be finely tuned [17]. Active ingredients with cosmetic efficacy can be vehiculated in the hydrotalcite, creating an intercalation compound with improved properties [18,19]. Moreover, the intercalation compound can release the stored active in a controlled way. To the best of our knowledge, only one article on the intercalation of caffeic acid on a LDH (MgAl-NO$_3$) has been published [20].

The purpose of this work is to create a smart active ingredient able to vehiculate caffeic acid in order to obtain higher local bioavailability and a prolonged anti-wrinkle effect. The ZnAl-LDH was chosen as host material for this purpose. In this article, the successful intercalation of caffeic acid into the interlayer region of an LDH is reported. The intercalation process is expected to enhance the antioxidant activity and the bioavailability and stability of caffeic acid, thus allowing its use in a broad spectrum of cosmetic applications. A deep chemical-physical and functional characterization of the ZnAl-LDH hybrid with caffeate anions is reported. The release studies and the antioxidant activities of the new hybrid compared to the free caffeic acid are also determined.

## 2. Materials and Methods

### 2.1. Chemicals and Solvent

$Zn(NO_3)_2\cdot 6H_2O$ and $CH_3COONa\cdot 3H_2O$ were purchased by Alfa Aesar (Haverhill, MA, USA). $Al(NO_3)_3\cdot 9H_2O$, caffeic acid $C_9H_8O_4$ and 2,2-Diphenyl-1-picrylhydrazyl (DPPH), monobasic potassium phosphate ($KH_2PO_4$) and bibasic sodium phosphate heptahydrate ($Na_2HPO_4\cdot 7H_2O$) were furnished from Sigma-Aldrich (Saint Louis, MO, USA). NaOH 1N was purchased from Carlo Erba (Cornaredo, Milano, Italy). Ethanol was provided from Applichem Panreac (Darmstadt, Germany). The solvents and materials were of reagent grade and were used without further purification.

### 2.2. ZnAl-ACE Preparation

With the aim of facilitating the vehiculation of caffeic acid, a proper LDH-layered matrix intercalated with acetate anions was prepared. The ZnAl-LDH in acetate form (ZnAl-ACE) was synthesized by coprecipitation method. 59.49 g of $Zn(NO_3)_2 \cdot 6H_2O$, 31.9 g of $Al(NO_3)_3 \cdot 9H_2O$, 193.98 g of $CH_3COONa_x \cdot 3H_2O$ were dissolved in final volume of 400 mL of deionized water. NaOH 1 N was added to the prepared solution until a pH value of 8 was achieved. The as-obtained precipitate was washed twice with deionized water and dried in oven at 60 °C.

### 2.3. ZnAl-CAF Preparation

Caffeic acid was salified by addiction of a stoichiometric amount of NaOH 1 N solution. The as-obtained salt was dissolved in a mixture ethanol/water 1:1 in order to obtain a concentration of 0.1 M. The ZnAl-ACE was equilibrated with the caffeate solution (1 g ZnAl-ACE/28 mL of solution) under magnetic stirring for 24 h. The intercalation was carried out at room temperature in $N_2$ atmosphere and protecting the sample from light. The resulting compound—labeledled as ZnAl-CAF—was recovered by centrifugation, washed twice with deionized water/ethanol 1:1 and dried in oven at 30 °C. The obtained compound was stored at 5 °C.

### 2.4. Sample Characterization

#### 2.4.1. Inductively Coupled Plasma-Optical Emission Spectrometer (ICP-OPES)

The Zn and Al content in the solid ZnAl-ACE was determined by ICP-OES Agilent Varian 700-ES series (Santa Clara, CA 95051, USA) after dissolving the sample in concentrated $HNO_3$ and properly diluting it.

#### 2.4.2. Thermal Analysis

Thermogravimetric analyses (TGA) of the sample ZnAl-ACE was carried out with an STD Q600 thermal analyzer TA Instrument (Luken Dr, New Castle, DE, USA) in air flow with a heating rate of 10 °C/min.

#### 2.4.3. UV-Visible Double-Beam Spectrophotometer

UV-visible analyses were carried out by UV-visible double-beam spectrophotometer (Jasco V-750, Oklahoma City, OK, USA). Measurements on powder samples were collected using a diffuse reflectance sphere accessory. As reference, a physical mixture (labeled as Phys. Mix.) between sodium caffeate and ZnAl-ACE was also prepared and characterized. The spectra were recorded on the solid ZnAl-ACE, ZnAl-CAF, Nacaffeate, and Phys. Mix. samples without dilution.

#### 2.4.4. FT-IR Analysis

FT-IR spectra were recorded at room temperature using a FT-IR-ATR Nicolet 380 (Thermo Fisher, Waltham, MA, USA). Typically, each spectrum was obtained in the spectral region from 400 to 4000 cm$^{-1}$. For the data collection, an attenuated total reflection crystal in SeZn was used.

#### 2.4.5. Caffeic Acid Content in ZnAl-CAF

The caffeic acid content in the sample ZnAl-CAF was determined by UV-Vis spectroscopy. A weighed amount of the sample (~20 mg) was completely dissolved in 100 mL of a 0.85 M aqueous solution of HCl. The caffeic acid content in the sample was determined by monitoring the maximum absorption value of the molecule at λ = 323 nm after calibration and proper dilution (dilution 1:5).

A suitable calibration was carried out by dissolving known amounts of pure caffeic acid in a proper volume of a 0.85 M aqueous solution of HCl. Five standards and a blank sample were then prepared and analyzed in order to obtain the calibration curve. The correlation coefficient of the calibration r was 0.9995 (Figure 2).

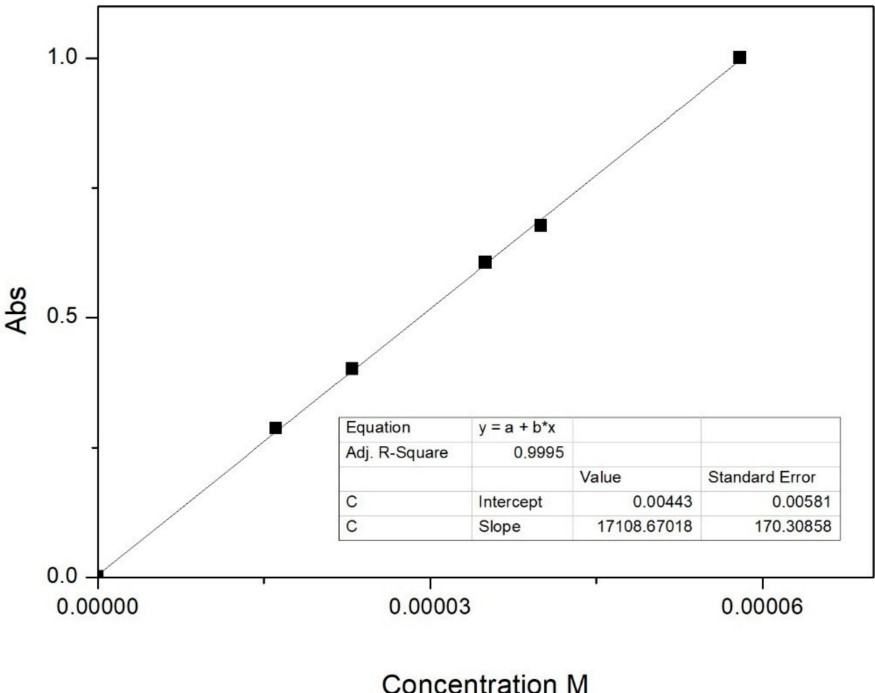

**Figure 2.** Linear regression for the determination of caffeic acid in a 0.85 M aqueous solution of HCl.

### 2.4.6. X-ray Powder Diffraction

The X-ray powder diffraction (XRPD) patterns of both sample were recorded with a Bruker D2 Phaser diffractometer (Billerica, MA, USA) operating at 30 kV and 15 mA, a step size 0.02 $2\theta$ degrees, and time per step of 1 s using the Cu K$\alpha$ radiation and a multistrip LYNXEYE SSD160 detector.

### 2.4.7. Franz Cell Release

Active release studies from pure caffeic acid and ZnAl-CAF were performed using a Franz Diffusion Cell (PermeGear, Inc., Bethlehem, PA, USA, 20 mm diameter) equilibrated at 32 °C by an external circle bath. The Franz Cell is composed of a donor chamber and a receiver chamber with a volume of 15 mL. These two parts were separated by a circular cellulose membrane (Filter paper Whatman 41, 20–25 mm, Whatman GmbH, Dassel, Germany). Phosphate buffer at pH 5.5 (F.U. XII ed., Italian Official Pharmacopoeia) was chosen as acceptor solution in order to mimic the skin composition. The donor solution was a solution of sodium carbonate (0.025 N) simulating the air composition. The amount of samples (caffeic acid, ZnAl-CAF) used in the test were calculated in order to ensure "sink" conditions indicated in the United States Pharmacopoeia. Pure caffeic acid (13 mg) and ZnAl-CAF (46.2 mg, corresponding to 13 mg of vehiculated caffeic acid) were dispersed in the donor chamber solution. Aliquots (0.4 mL) of the acceptor fluid were collected at predetermined time intervals (5, 10, 20, 30, 60, 120, 180, 240, 300, and 360 min) and immediately replaced by the same volume of fluid equilibrated at 32 °C. The withdrawn samples were properly diluted. Tests were carried out in triplicate, and the results were reported as average values. The released acid was detected by UV-Vis spectrophotometry at the wavelength of maximum absorption of caffeic acid in the phosphate buffer ($\lambda$ = 312 nm) using an UV-visible double-beam spectrophotometer. A suitable calibration was carried out by dissolving known amounts of pure caffeic acid in a proper volume of phosphate buffer. Five standards and a blank sample were then prepared and analyzed in order to obtain the calibration curve. The correlation coefficient of the calibration r was 0.9994 (Figure 3).

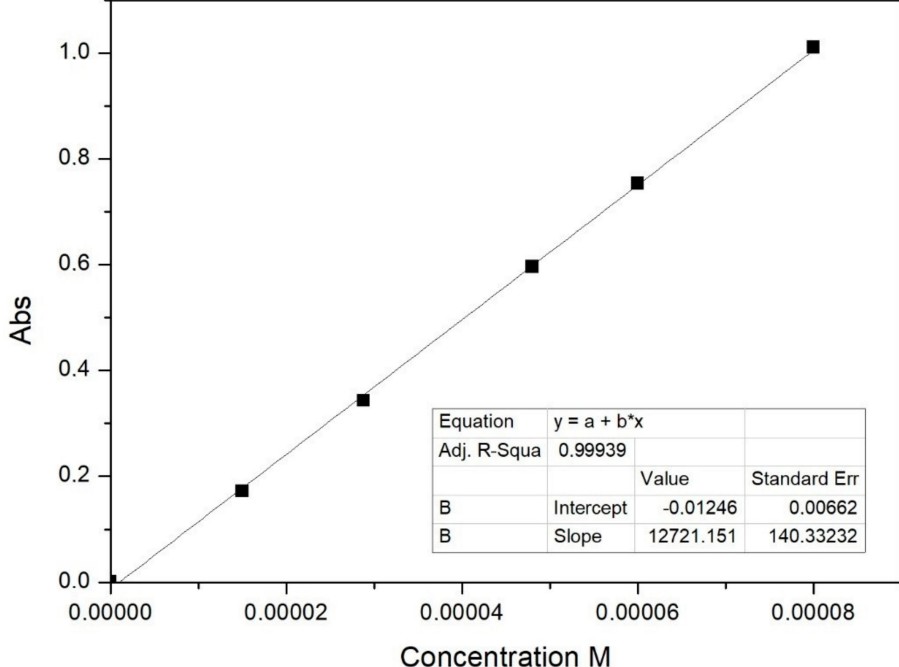

**Figure 3.** Linear regression for the determination of caffeic acid in the phosphate buffer.

### 2.4.8. Antioxidant Activity

Antioxidant activities of ZnAl-CAF and of caffeic acid were evaluated using the DPPH free radical method [21]. When DPPH reacts with an antioxidant compound able to donate a hydrogen atom, the radical on DPPH is reduced and its solution consequently changes color from violet to pale yellow. The resulting color change is proportional to the number of electrons captured and can be related to the antioxidant activity. A $6 \times 10^{-5}$ M solution of DPPH in a 1:1 ethanol/phosphate buffer at pH 5.5 (F.U.) mixture was prepared. One liter of this solution was magnetically stirred, protected from light. The absorbance of DPPH at the wavelength of maximum absorption in this solvent was recorded ($\lambda$ = 525 nm). This value of absorbance was considered as 100% of DPPH in solution. Caffeic acid (in caffeic acid/DPPH molar ratio of 0.25) was added and the change of color monitored by collecting aliquots at predetermined time and analyzing them using the UV-Vis spectrophotometer. The experiment was repeated with the sample ZnAl-CAF maintaining the indicated molar ratio of antioxidant molecule/DPPH (corresponding to 9.6 mg of ZnAl–CAF). A control experiment was conducted using the same amount (9.6 mg) of ZnAl-ACE.

## 3. Results and Discussion

### 3.1. LDHs Characterization

The LDH was directly synthesized in acetate form by the coprecipitation method [22]. The ZnAl-ACE composition was determined by ICP-OES and TGA analyses and it is hereinafter reported: $[Zn_{0.68}Al_{0.32}(OH)_2](CH_3COO)_{0.32} \cdot 0.5H_2O$ ZnAl-ACE was used as starting material to intercalate caffeic acid. The acetate form as pristine material was chosen, after several failed intercalation attempts, in order to facilitate the ion exchange process. The intercalation was carried out at room temperature with a green procedure that was easy to scale up. XRPD analysis of the samples ZnAl-ACE and ZnAl-CAF are reported in Figure 4. The pristine material had an interlayer distance of 12.6 Å, which could be assigned to the presence of the acetate anions in the interlayer region. The interlayer distance in ZnAl-CAF increased from 12.6 Å to 16.3 Å, confirming the intercalation. These values were compatible with the intercalation of the caffeate and in good agreement with

data reported in literature for the intercalation of the ferulate, which possesses a similar chemical structure [23].

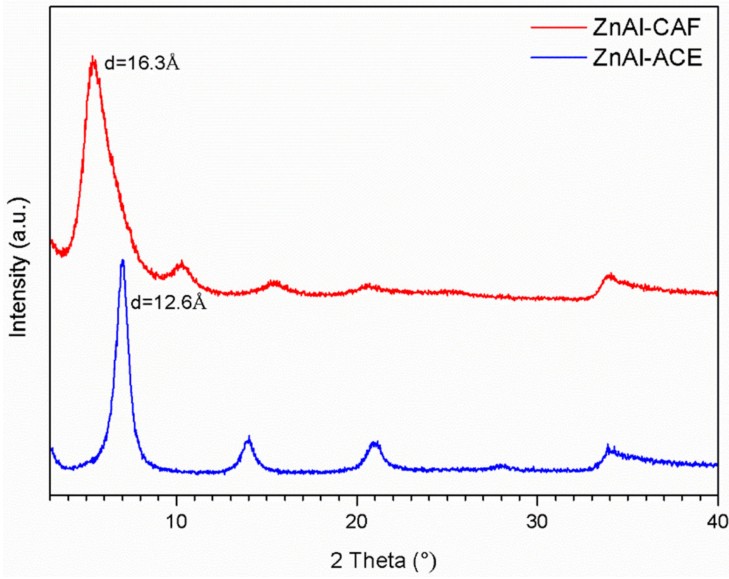

**Figure 4.** XRPD spectra of the ZnAl-ACE (blue line) and ZnAl–CAF (red line) samples.

The caffeate content was determined by dissolving the sample in a solution of HCl and analyzing it by UV-Vis spectroscopy. The loading was $28.1 \pm 0.2$ $w/w\%$, corresponding to the following formula: $[Zn_{0.68}Al_{0.32}(OH)_2](CAF)_{0.27}(CH_3COO)_{0.05} \cdot 2H_2O$. From the determined chemical formula, it is evident that some acetate anions were still present in the interlayer region. According to the chemical composition, the experimental active loading corresponded to about 84 wt% of the total caffeic acid used in the intercalation process, indicating a good yield of the ion exchange reaction. UV-Vis reflectance spectra of ZnAl-ACE, ZnAl-CAF, sodium caffeate, and the physical mixture between ZnAl-ACE and sodium caffeate (Phys. Mix.) are reported in Figure 5. The Phys. Mix. sample was prepared according to the experimental active loading observed for the ZnAl-CAF (i.e., 71.9% of ZnAl-ACE and 28.1% of caffeic acid). Recorded UV-Vis spectra confirmed the effective intercalation of the caffeate in the LDH. The spectrum of ZnAl-CAF showed an increase in intensity and in the absorption range, both in the UV and in the visible regions, as already described in the literature [24]. These data indicate that the synthesized inorganic-organic hybrid material possessed higher UV absorption capacity compared with both the pure caffeate and the physical mixture between ZnAl-ACE and caffeate. This result can open new and interesting applications in the field of photoprotection of UV-sensible actives. Similar results and considerations have been obtained for the intercalation of other active compounds in LDHs [23,25].

In Figure 6, the comparison between the FT-IR spectra of ZnAl-CAF, pristine LDH, pure sodium caffeate, and Phys. Mix. samples are reported. The broad absorption band between 3500 and 3000 $cm^{-1}$ in ZnAl-ACE, ZnAl-CAF, and Phys. Mix. samples could be assigned to the OH vibrations of layer hydroxyls and interlayer water [23]. Sodium caffeate possessed some narrow bands in the region between 1700 and 800 $cm^{-1}$ due to the vibrational mode of the C=C, C=O, COO$^-$ functional groups. The bands between 1515–1540 $cm^{-1}$ and 1395–1420 $cm^{-1}$ were associated to the symmetric and asymmetric stretchings of COO$^-$, respectively [26]. The bands of caffeate were also present in the spectrum of ZnAl-CAF and Phys. Mix. Samples but absent in ZnAl-ACE, confirming the presence of the active in the intercalation compound. Some of the peaks of caffeate on ZnAl-CAF were at the same frequency, while others were slightly shifted. This shift indicated that caffeate was intercalated in the LDH [20].

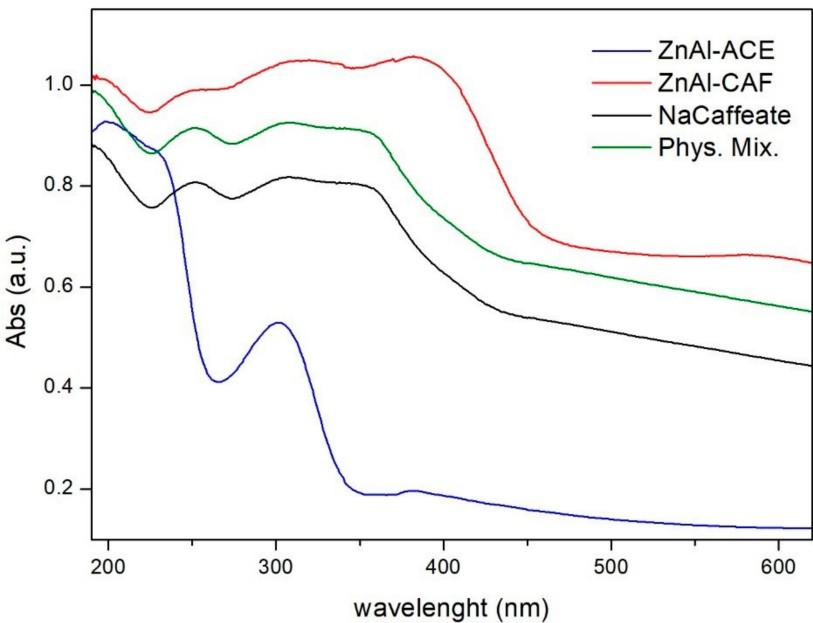

**Figure 5.** UV-Vis spectra of the ZnAl-ACE (blue line), ZnAl-CAF (red line), NaCaffeate (black line), Phys. Mix. (green line) samples.

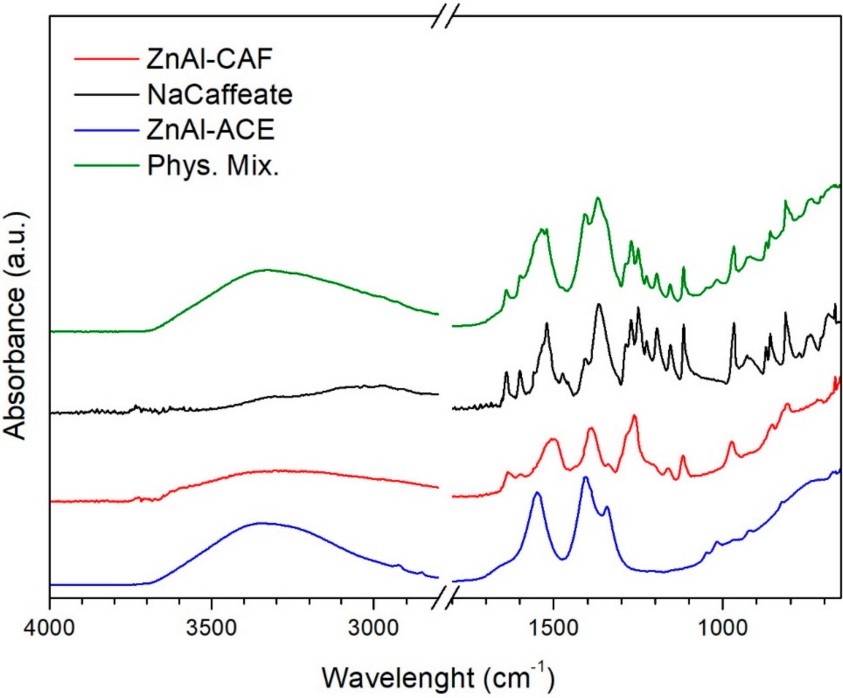

**Figure 6.** FT-IR spectra of the ZnAl-ACE (blue line), ZnAl-CAF (red line), NaCaffeate (black line), Phys. Mix. (green line) samples.

### 3.2. In Vitro Release Studies

The in vitro release studies of the ZnAl-CAF and the pure active were performed using the Franz Diffusion Cell. Caffeic acid was chosen as reference for this test and for the antioxidant activity evaluation because it is the standard raw material with a registered INCI (International Nomenclature of Cosmetic Ingredients) name currently used in the cosmetic market. This system of release was chosen because it is considered the most appropriate for simulating topical application [27]. All release

experiments showed a good reproducibility; the release profiles are reported in Figure 7. The results are expressed as % of the released active in the acceptor solution referred to the total active content in the donor cells. The ZnAl-CAF sample showed a higher active release during the whole analyzed time range compared to the pure caffeic acid. After 6 h, the release from the pure caffeic acid reached 59% of the total active, while ZnAl-CAF reached a value of 78%. This result clearly indicated that the innovative hybrid had a greater bioavailability than the pure active. In fact, thanks to the intercalation technology, the active species was hosted in the interlayer region of the LDH in a "liquid like" state that enhanced their dissolution rate. In these conditions, when the anionic exchange was activated, the caffeic acid was directly released in the medium, bypassing the crystal dissolution stage [28,29].

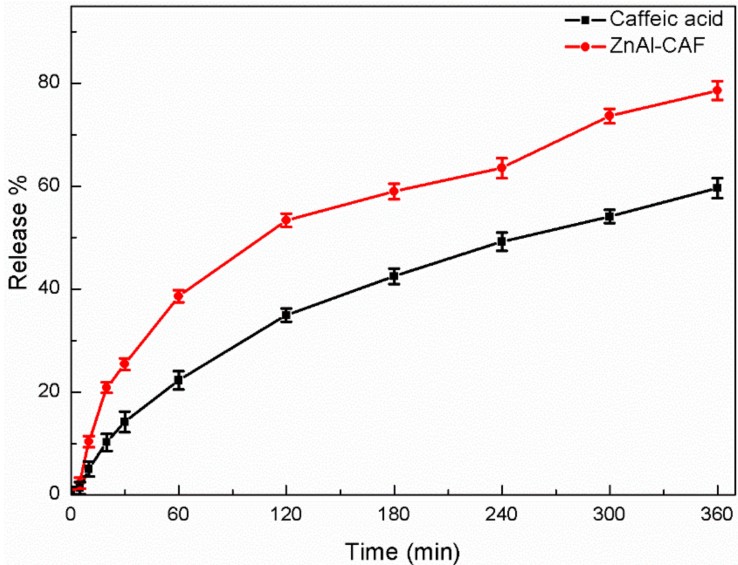

**Figure 7.** Release curves of the active from the pure caffeic acid (black line) and the ZnAl-CAF (red line) sample.

### 3.3. Antioxidant Activity

In order to evaluate the antioxidant properties of ZnAl-CAF compared to the pure active and to the pristine LDH, the DPPH method was chosen. The experiments were performed in an excess of DPPH to exhaust the H-donating capacity of the antioxidant [21]. Due to DPPH solubility, the test is typically conducted in ethanol or methanol. In this case, the test was carried out in a 1:1 ethanol/phosphate buffer mixture to better simulate the antioxidant behavior in a topical application. In Figure 8, the % of remaining DPPH in the solution as a function of time (min) is reported. As shown in the graph, the antioxidant activity of the pure caffeic acid immediately expired, and after only one minute, the percentage of DPPH in solution decreased up to the 10% and remained unchanged for the whole test. The pristine ZnAl-ACE (blue line), used as control experiment, showed no activity in the DPPH reduction. These data confirmed that the antioxidant properties were attributable only to the intercalation of caffeic acid into the interlayer region.

The ZnAl-CAF sample showed a slower and gradual antioxidant activity; this has already been demonstrated for other intercalated antioxidants, such as gallic acid and carnosine [30]. The intercalation technology protected the caffeic acid from the oxidant environment and the slow release activated by the presence of anions, allowing the realization of formulation with prolonged antioxidant activity.

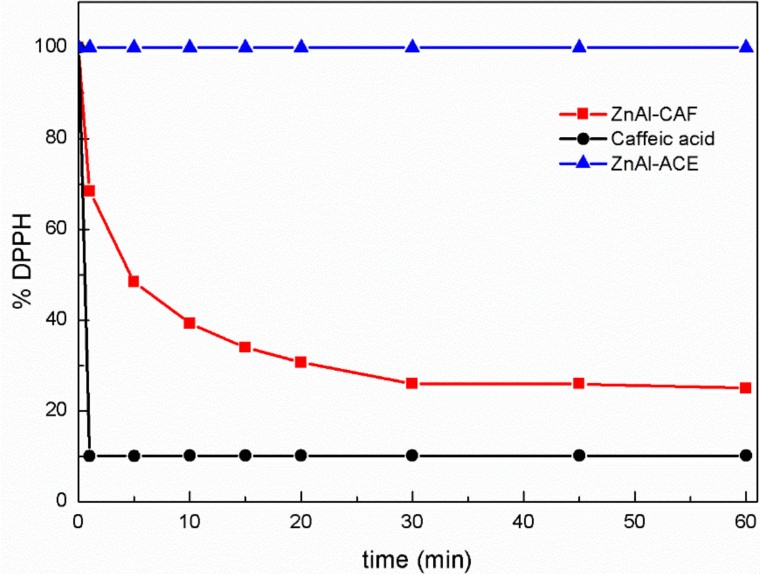

**Figure 8.** Antioxidant activities of the pure caffeic acid (black line), the ZnAl-CAF (red line), and the ZnAl-ACE (blue line) sample.

## 4. Conclusions

In this article, the successful preparation of an innovative raw material for cosmetic application is reported. Caffeic acid was vehiculated in synthetic clay with a high loading, and the hybrid showed very promising properties. The Franz cell release test showed that ZnAl-CAF possessed an increased bioavailability compared with the neat caffeic acid, giving hope to a great improvement in its efficacy. The sustained antioxidant activity observed in the ZnAl-CAF demonstrated that the caffeic acid was protected from oxidation during the storage in the interlayer region. Furthermore, data showed that the function of caffeic acid could be possibly prolonged beyond 1 h after the application through the intercalation/vehiculation approach, whereas the pure free ingredient exhausted its antioxidant activity in the first minute. The ability of LDH to act as a very efficient carrier for active ingredients has been widely demonstrated in the literature [28,30]. Moreover, Cunha et al. [31] reported that ZnAl-LDH are also biocompatible, nontoxic and nonimmunogenic after in vivo tests. The intercalation technology can greatly increase the use of caffeic acid in the cosmetic market, enabling new effective dermo-cosmetic formulations for anti-wrinkle and antipollution applications.

**Author Contributions:** M.S., M.B. and A.P. designed the experiments. C.F. synthesized and chemically–physically characterized the samples. M.B. and A.P. carried out the release tests and the antioxidant activity studies.

**Funding:** This research received no external funding.

**Acknowledgments:** The authors would like to thank Catia Clementi from the University of Perugia for the scientific collaboration.

**Conflicts of Interest:** The authors declare no conflict of interest.

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
