# Peer review of "Caffeic Acid-layered Double Hydroxide Hybrid: A New Raw Material for Cosmetic Applications"

_cosmetics, doi:10.3390/cosmetics5030051_

Round 1

Reviewer 1 Report

Dear authors, the following  results must be improved:

1) in the UV-spectroscopy a linear regression curve and analysis of physical mixture between caffeic acid and LDL in order to evaluate the possible interference of the matrix in the analysis are missing. In fact in the Figure 3 the ZnAl-CAF profile can be only the sum of the two separate profiles and it can not confirm the effective intercalation.

2) same aspect has to be considered in the FT-IR analysis. Without the analysis of the physical mixture between the ZnAl-ACE and the caffeic acid you can not confirmed anything. 

3) in the diffusion test using Franz cells it is not clear if caffeic acid or its sodium salt is used as control. It is very important to highlight also the sensitivity of the method. Furthermore, if the  caffeic acid  dissolved of about 50% in 4 hours, it is already a slow release, What advantage could there be in introducing the active ingredient into the LDL? It is not clear at all.

4)About DPPH test: your results are not conclusive; in fact the antioxidant activity of ZnAl-CAF is tested alone. No data ie reported about the activity of ZnAl-ACE.  It could be important to test the antioxidant activity of ZnAl-CAF during the release test in the Franz cells to confirm the capability to prolong antioxidant activity.

Author Response

To the Editor of Cosmetics and to Reviewer 1

Perugia, August 2018

Dear Editor, dear reviewer 1,

Thank you for your kind attention in our work. We read the reviewer comments and we carefully modified the manuscript. We hope that our corrections can satisfy your requests, in order to publish the article in Cosmetics.

You can find enclosed a revised copy of the manuscript. The inserted corrections are highlighted in yellow. Hereafter we reported a clarification of each reported correction.

Reviewer 1

1) In the UV-spectroscopy a linear regression curve and analysis of physical mixture between caffeic acid and LDH in order to evaluate the possible interference of the matrix in the analysis are missing. In fact in the Figure 3 the ZnAl-CAF profile can be only the sum of the two separate profiles and it can not confirm the effective intercalation.

The UV-vis spectrum of the physical mixture has been inserted in the text and commented. This characterization is a qualitative analysis and no calibration curve was carried out. The linear regression curves were realized for the quantitative determination of the active loading in the ZnAl-CAF and the release tests. The linear regression curves together with the relative equations were reported in the text (Figure 2-3).

2) Same aspect has to be considered in the FT-IR analysis. Without the analysis of the physical mixture between the ZnAl-ACE and the caffeic acid you can not confirmed anything. 

The FT-IR spectrum of the physical mixture was inserted and commented in the text.

3) In the diffusion test using Franz cells it is not clear if caffeic acid or its sodium salt is used as control. It is very important to highlight also the sensitivity of the method. Furthermore, if the caffeic acid dissolved of about 50% in 4 hours, it is already a slow release, What advantage could there be in introducing the active ingredient into the LDH? It is not clear at all.

In materials and methods paragraph it is specified that caffeic acid was used as reference for the release test. The relative amounts of acid and caffeate forms of course depend on the pH of the release medium. In this work a pH of 5.5 was employed so in this case the two forms are both present due to the acid-base equilibrium. Regarding the sensitivity of the method, the linear regression curve used for the determination of the caffeic acid in the sample has been reported in the materials and methods section (Figure 3). About the behaviour of the sample ZnAl-CAF in the release test, the advantage is described in the text: the innovative hybrid has a greater bioavailability than the pure active, and not only a slow release.

4) About DPPH test: your results are not conclusive; in fact the antioxidant activity of ZnAl-CAF is tested alone. No data ie reported about the activity of ZnAl-ACE.  It could be important to test the antioxidant activity of ZnAl-CAF during the release test in the Franz cells to confirm the capability to prolong antioxidant activity.

- DPPH tests: the antioxidant activity on the control sample ZnAl-ACE has been inserted.

- Franz cell tests: Antioxidant activity is an intrinsic property of the molecule (as indicated by the bibliography in the introduction). Since the UV-vis spectra of the released caffeic acid showed the same profile of the pristine pure molecule and no degradation effects were detected, it is reasonable to suppose that the antioxidant activity is maintained after the release.

Yours sincerely,

Maria Bastianini

Reviewer 2 Report

It was my honor being invited to review the manuscript entitled "Caffeic acid-layered double hydroxide hybrid: a new raw material for cosmetic applications", regarding the vehiculation of caffeic acid in a layered double hydroxides, in order to increase its bioavailability and antioxidant activity. This technique seems to possess an interesting application in the cosmetic market. The study is well-structured, and the conclusions are supported well by the data. Moreover, the results provides important information about the intercalation technology, since the bibliography in this field is inadequate.

For the aforementioned reasons, I think that the manuscript should be published and can be further considered for publication in Cosmetics. Although, there are some minor concerns that must be revised and clarified in order to published.

>        In the “Introduction” part, please clarify the purpose of this study in a separate paragraph extensively.

>        In the “Materials and Methods” part, please refer the statistical analysis and parametric tests used.

>        In the “Conclusions” part: A) Please discus more extensively your results. B) You should also, make a reference in another bibliographic data, which have already used the intercalation technology, and compare your findings. C) Is there any bibliographic data which confirms that these intercalated products have no allergic reactions in vivo, in order to use as cosmetics?

>        In the “References” part, please replace the references with the numbers 3, 5, 7, 8, 20, 21 with more recent bibliographic data.

Author Response

To the Editor of Cosmetics and to Reviewer 2

Perugia, August 2018

Dear Editor, dear reviewer 2,

Thank you for your kind attention in our work. We read the reviewer comments and we carefully modified the manuscript. We hope that our corrections can satisfy your requests, in order to publish the article in Cosmetics.

You can find enclosed a revised copy of the manuscript. The inserted corrections are highlighted in yellow. Hereafter we reported a clarification of each reported correction.

Reviewer 2

1) In the “Introduction” part, please clarify the purpose of this study in a separate paragraph extensively.

The requested paragraph was inserted in the “Introduction”.

2) In the “Materials and Methods” part, please refer the statistical analysis and parametric tests used.

More detailed information about the tests were added in the experimental part. Linear regression curves together with the relative equation for the UV-vis quantitative analysis (active loading determination in the ZnAl-CAF and release test) were inserted in the text (Figure 2-3).

3) In the “Conclusions” part: A) Please discus more extensively your results. B) You should also, make a reference in another bibliographic data, which have already used the intercalation technology, and compare your findings. C) Is there any bibliographic data which confirms that these intercalated products have no allergic reactions in vivo, in order to use as cosmetics?

The “Conclusion” part was expanded with extra comments and bibliographic data.

4) In the “References” part, please replace the references with the numbers 3, 5, 7, 8, 20, 21 with more recent bibliographic data.

The references were replaced, except for the last one, that is the only review on the preparation of LDH in acetate form.

Yours sincerely,

Maria Bastianini

Reviewer 3 Report

Although extensive editing of English language and style is required, the results of the paper by Bastianini et al. seem interesting and worth to be published. However, some points should be addressed:

1)      The abstract should be improved and include the main findings of the study

2)      Page 2, line 50: authors should better explain the general formula of LDHs. In particular, what does “x” stand for? Is it always <1? Otherwise “1-x” would be a negative number…

3)      Page 2, line 57: authors should introduce the structure of hydrotalcite

4)      Page 2, line 67: please remove the dot near DPPH

5)      Page 2, line 72: please define “ACE”

6)      Page3, line 114: please define “F.U.”

7)      Page 6, lines 116-118: the concentration of the samples in the release experiments should be indicated. In particular, was the molar concentration of CAF in the experiments run with pure caffeic acid and ZnAl-CAF the same? Otherwise, the higher release observed with the hybrid is not meaningful.

8)      Page 4, line 153: authors should report the percentage of loading as a mean + SD value. Moreover, how much of the used caffeic acid was incorporated?

9)      Figure 3: what is the concentration of the samples used to record the UV-vis spectra? And what about the broad band centered at ca. 580 nm that characterizes the ZnAl-CAF sample? I suggest performing HPLC analysis to check that CAF does not undergo any chemical transformations during intercalation

10)  Figure 5 and Figure 6: why did authors use caffeic acid and not its sodium salt?

11)  Figure 6: authors should perform and show control experiments on ZnAl-ACE

Author Response

To the Editor of Cosmetics and to Reviewer 3

Perugia, August 2018

Dear Editor, dear reviewer 3,

Thank you for your kind attention in our work. We read the reviewer comments and we carefully modified the manuscript. We hope that our corrections can satisfy your requests, in order to publish the article in Cosmetics.

You can find enclosed a revised copy of the manuscript. The inserted corrections are highlighted in yellow. Hereafter we reported a clarification of each reported correction.

Reviewer 3

1) The abstract should be improved and include the main findings of the study

The abstract was modified according to the request.

2) Page 2, line 50: authors should better explain the general formula of LDHs. In particular, what does “x” stand for? Is it always <1? Otherwise “1-x” would be a negative number…

This point was clarified, explaining “x” meaning.

3) Page 2, line 57: authors should introduce the structure of hydrotalcite

The structure of hydrotalcite was briefly explained and a reference has been inserted.

4) Page 2, line 67: please remove the dot near DPPH

The dot was removed.

5) Page 2, line 72: please define “ACE”

“ACE” is referred to the intercalated acetate anions; the definition was inserted in the text.

6) Page3, line 114: please define “F.U.”

“F.U.” was defined in the text.

7) Page 6, lines 116-118: the concentration of the samples in the release experiments should be indicated. In particular, was the molar concentration of CAF in the experiments run with pure caffeic acid and ZnAl-CAF the same? Otherwise, the higher release observed with the hybrid is not meaningful.

The amount of caffeic acid inserted in the donor cell is the same for the reference and ZnAl-CAF, and was calculated considering the experimental active loading observed in the hybrid. The relative amount of each sample (i.e. also the concentration because the volume of the cell is known and indicated in the text) used for the test was inserted in “Materials and methods” paragraph.

8) Page 4, line 153: authors should report the percentage of loading as a mean + SD value. Moreover, how much of the used caffeic acid was incorporated?

The final compound ZnAl-CAF contained 28.1 ± 0.2 w/w% of active. According to the chemical formula [Zn0,68Al0,32(OH)2](CAF)0,27(CH3COO)0,05 · 2H2O, the intercalated caffeic acid corresponds to about 84 wt% of the total employed amount of caffeic acid. This observation was highlighted in the text.

9) Figure 3: what is the concentration of the samples used to record the UV-vis spectra? And what about the broad band centered at ca. 580 nm that characterizes the ZnAl-CAF sample? I suggest performing HPLC analysis to check that CAF does not undergo any chemical transformations during intercalation

The UV-vis spectra were recorded on the solid samples without dilution. In particular, ZnAl-ACE, ZnAl-CAF, Nacaffeate and the Physical Mixture (with the same ZnAl-ACE/caffeic acid ratio of the hybrid) were reported. This detail was specified in the “Materials and methods” paragraph. The content of caffeic acid was determined by UV-vis spectroscopy using the linear regression reported in Figure 2.

10) Figure 5 and Figure 6: why did authors use caffeic acid and not its sodium salt?

The caffeic acid was chosen as reference in the functional tests (release tests and antioxidant activities) because it is the standard competitor of our innovative proposal. It has a registered INCI name and it is already used in the cosmetic field. Of course the relative amounts of acid and caffeate forms depend on the pH of the medium and at certain pH values the two forms can coexist. A brief clarification was added in the text.

11) Figure 6: authors should perform and show control experiments on ZnAl-ACE

The control experiment was inserted and commented.

Yours sincerely,

Maria Bastianini
